# Finding a Jill for JAK: Assessing Past, Present, and Future JAK Inhibitor Combination Approaches in Myelofibrosis

**DOI:** 10.3390/cancers12082278

**Published:** 2020-08-14

**Authors:** Andrew T. Kuykendall, Nathan P. Horvat, Garima Pandey, Rami Komrokji, Gary W. Reuther

**Affiliations:** 1Department of Malignant Hematology, H. Lee Moffitt Cancer Center & Research Institute, Tampa, FL 33612, USA; Rami.Komrokji@moffitt.org; 2Morsani College of Medicine, University of South Florida, Tampa, FL 33612 USA; nhorvat@usf.edu; 3Department of Molecular Oncology, H. Lee Moffitt Cancer Center & Research Institute, Tampa, FL 33612, USA; Garima.Pandey@moffitt.org (G.P.); Gary.Reuther@moffitt.org (G.W.R.)

**Keywords:** myelofibrosis, myeloproliferative neoplasm, JAK inhibitor

## Abstract

Myelofibrosis (MF) is a myeloproliferative neoplasm hallmarked by the upregulation of the Janus kinase (JAK)—signal transducer and activator of transcription (STAT) pathway with associated extramedullary hematopoiesis and a high burden of disease-related symptoms. While JAK inhibitor therapy is central to the management of MF, it is not without limitations. In an effort to improve treatment for MF patients, there have been significant efforts to identify combination strategies that build upon the substantial benefits of JAK inhibition. Early efforts to combine agents with additive therapeutic profiles have given way to rationally designed combinations hoping to demonstrate clinical synergism and modify the underlying disease. In this article, we review the preclinical basis and existing clinical data for JAK inhibitor combination strategies while highlighting emerging strategies of particular interest.

## 1. Introduction

Myelofibrosis (MF) is a BCR-ABL1-negative myeloproliferative neoplasm (MPN) with heterogeneous clinical features that include high and low blood counts, splenomegaly, and constitutional symptoms [1]. The vast majority of MF patients harbor a phenotype-driving activating mutation in one of three genes—*JAK2*, *MPL*, and *CALR*—all of which lead to upregulation of the JAK/STAT pathway [2]. Ruxolitinib and fedratinib are oral medications that inhibit mutant and wild-type Janus kinase-2 (JAK2) and block the hyperactive JAK/STAT signaling, which is a hallmark of MPNs. Both are approved by the United States Food and Drug Administration (FDA) for the treatment of intermediate and high-risk MF due to their ability to reduce spleen volume and improve disease-related symptoms [3,4,5]. Despite demonstrating clinical benefit, JAK2 inhibitors have several drawbacks. First, despite being used in patients who frequently exhibit anemia and thrombocytopenia, JAK2 inhibitors often worsen hematologic parameters in a dose-dependent manner, limiting their use in a substantial proportion of patients. Additionally, JAK2 inhibitors have not shown the consistent ability to modify the underlying disease, failing to induce molecular remissions or prevent disease progression [6,7,8]. In light of this, recent drug development in MF has largely focused on finding the appropriate combination partner to build upon the successes of JAK inhibition. In this review, we aim to review past and current combination strategies in MF, combining the preclinical rationale with the available clinical data and highlighting emerging targets for combination approaches. The primary focus of this review will be on ruxolitinib-based combinations, but assessment of fedratinib-based combinations will surely be initiated in a short time given its relatively recent FDA approval.

## 2. Rationale for Combination Therapy

In general, combination strategies either attempt to achieve multiple endpoints that cannot be achieved with an individual agent or achieve a single endpoint more effectively than is possible with a single agent. Pursuance of a combination approach implies that the existing single-agent approach has benefits and limitations. Ruxolitinib was approved for intermediate- and high-risk MF based on two phase 3 clinical trials that demonstrated an ability to reduce spleen size and improve disease-related symptoms. In the COMFORT-I study, 41.9% of ruxolitinib-treated patient experienced ≥35% reduction in spleen response at 24 weeks compared to 0.7% of patients treated with placebo. Additionally, 45.9% of ruxolitinib-treated patients experienced 50% reduction in total symptom compared to 5.3% of patients in the placebo arm [4]. In the COMFORT-II study, ruxolitinib was compared to the best available therapy (BAT) and demonstrated superior spleen volume reduction at week 24 and 48 with 32% and 28% achieving at least 35% reduction in spleen volume, respectively, compared to 0% of patients treated with BAT at both time points. Notably, ruxolitinib-treated patients experienced improvement in patient-reported outcomes based on several different measures [3]. Beyond an impact on spleen size and disease-related symptoms, ruxolitinib has shown favorable impact on nutritional parameters and hepatomegaly, while its impact on JAK2 allele burden and bone marrow fibrosis has been modest, with more significant potential impact emerging with long-term treatment [7,9,10,11,12]. Perhaps due to these favorable impacts, ruxolitinib has been associated with a survival benefit in independent and pooled analyses of the COMFORT trials [6,13,14,15].

Nevertheless, ruxolitinib therapy is not without limitations. A majority of patients will discontinue treatment by three years due in equal parts to disease progression, adverse events, or death [16,17]. Ruxolitinib was associated with grade ≥3 anemia in 45% and 42% of patients and grade ≥3 thrombocytopenia in 13% and 8% of patients on the COMFORT-I and COMFORT-II trial, respectively [3,4]. Additionally, due to impacts on innate and adaptive immunity, ruxolitinib is associated with an increased risk of infectious complications, such as herpes zoster reactivations, and nonmelanoma skin cancers [6,18,19]. The presence of such risks reinforces the fact that ruxolitinib is not an ideal treatment option for MF patients who would otherwise not have been included on the COMFORT studies, specifically those with lower-risk disease, thrombocytopenia, and without significant splenomegaly.

A detailed analysis of ruxolitinib response data has provided an improved understanding of patients likely to derive the greatest benefit. Higher ruxolitinib doses, lower-risk disease, transfusion independency, lack of thrombocytopenia, and shorter duration of disease have been associated with more favorable spleen responses to ruxolitinib [20,21]. Clonal dynamics play a role as well, with higher JAK2 mutant allele fractions and lower number of mutations predicting favorable responses [22,23]. In addition, *RAS* pathway mutations have recently been associated with resistance to JAK inhibitors and poor outcomes [24,25,26]. Durability of spleen response to ruxolitinib treatment is associated with improved overall survival, while thrombocytopenia and clonal evolution correlate with inferior outcomes [27,28].

Building upon these data, combination approaches in MF aim to preserve the favorable impact ruxolitinib has on spleen volume, patient-reported outcomes, and overall survival while improving care for those patients who are not optimal candidates for ruxolitinib or those at risk of not experiencing the maximum benefit from therapy. Past and ongoing clinical studies of ruxolitinib combination therapy studies in myelofibrosis patients are highlighted in Table 1.

## 3. Killing Two Birds with Two Stones: Efforts to Manage Anemia while Controlling Splenomegaly/Symptoms with Ruxolitinib Therapy

### 3.1. Erythropoiesis Stimulating Agents (ESAs), Androgens (Danazol), and Iron Chelation Therapy (Deferasirox)

Due to the largely suppressive effects of JAK inhibitors on hematologic parameters, initial combination efforts focused primarily on MF patients with symptomatic anemia. In a post-hoc analysis, McMullin et al. reviewed the use of ESAs in patients receiving ruxolitinib in the COMFORT-II trial and demonstrated the combination was well tolerated, though rarely associated with a substantial benefit [51]. A subsequent retrospective analysis suggested a more favorable impact of adding ESA to ruxolitinib, with anemia responses occurring in 54% of patients, though the relative timing of ESA initiation to ruxolitinib varied [52]. While the concept of augmenting red blood cell production via stimulation of erythropoietin (EPO) receptors that are being blocked by downstream JAK2 inhibition seems illogical, ruxolitinib does not result in continuous JAK2 inhibition. The half-life of ruxolitinib in plasma is about 2.3 h, which, even with twice daily dosing, allows the EPO receptor to be stimulated by ESAs during times in which JAK2 is presumably suboptimally inhibited [53].

Along similar lines, Gowin et al. performed a multicenter phase 2 study assessing danazol in combination with ruxolitinib. Danazol, a semisynthetic attenuated androgen, has previously demonstrated the ability to improve anemia in MF patients as a single agent, with a response rate of 30% [54]. The combination was well tolerated; however, there were limited objective responses. Stable or increasing hemoglobin levels were noted in the majority of patients, along with a favorable impact on platelet counts. This study included patients who were newly starting ruxolitinib and would have otherwise been expected to exhibit a decrease in hemoglobin within the first two months of treatment, suggesting that the benefit of concomitant danazol could have been underestimated [29].

More recently, the impact of iron chelation therapy (ICT) in combination with ruxolitinib has been assessed. In a retrospective, multicenter study, 59 patients received ruxolitinib in combination with defersirox for the management of iron overload. Iron chelation response was achieved in 24 (41%) patients and 10 (17%) converted from transfusion dependence to transfusion independence [55].

### 3.2. Immunomodulatory Imide Agents (IMiDs)

Combination strategies with IMiDs, lenalidomide, thalidomide, and pomalidomide have also been pursued. As single agents, IMiDs have shown mild to moderate efficacy in improving anemia, thrombocytopenia, and splenomegaly in patients with MF [30,56,57,58,59,60,61,62,63,64,65,66]. In a prospective, single-center, phase II trial, Daver et al. evaluated the combination of ruxolitinib and lenalidomide in 31 patients with MF. This combination proved difficult as 74% of patients required dose interruption, which typically occurred within the first three months and was often due to cytopenias, particularly thrombocytopenia. Fourteen patients permanently discontinued lenalidomide within three months. Among 17 (55%) patients who achieved IWG-MRT-defined responses, all were due to spleen volume improvement, with only one patient experiencing clinical improvement in hemoglobin. Despite poor tolerability, reduction in *JAK2* V617F mutant allele burden was noted in 5/6 responding patients who had serial measurements, suggesting the potential for disease modification, although fibrosis improvement was rare.

The combination of ruxolitinib and pomalidomide is being studied in the ongoing MPNSG-0212 trial (NCT01644110), with the most recent update presented by Stegelmann et al. at the American Society of Hematology (ASH) Annual Meeting in 2019. In this multicenter, single-arm phase Ib/II study, a two-stage design is being used, wherein the first 40 patients receive ruxolitinib 10 mg twice daily with low-dose pomalidomide (0.5 mg daily), while the next 50 patients receive ruxolitinib along with a stepwise dose escalation of pomalidomide from 0.5 to 2 mg daily. In the fully accrued cohort 1, an objective anemia response was seen in 7/40 (18%) patients, with an additional 11 (27%) patients meeting study-defined criteria for clinical benefit. Enrollment of the second cohort receiving escalating doses of pomalidomide is ongoing [31].

The combination of ruxolitinib and thalidomide is also being investigated in an ongoing phase 2 study (NCT03069326). In this trial, patients received at least three months of ruxolitinib before the addition of thalidomide. In the most recent update from the 2019 ASH Annual Meeting, 23 of a planned 25 patients had been accrued, with 15 patients evaluable for response. Favorable impacts on anemia, thrombocytopenia, and spleen volume were observed in 6/8 (75%) patients, with baseline thrombocytopenia achieving a study-defined platelet response [32].

### 3.3. TGF-β Signaling Agents

Luspatercept and sotatercept are activin receptor II ligand traps that improve late-stage erythroblast differentiation by binding to TGFβ superfamily ligands and decreasing SMAD2 and SMAD3 signaling [67]. Luspatercept is FDA-approved for transfusion-dependent β-thalassemia, very low- to intermediate-risk myelodysplastic syndromes with ring sideroblasts (MDS-RS) and myelodysplastic/myeloproliferative neoplasm with ring sideroblasts and thrombocytosis (MDS/MPN-RS-T) [68]. Due to the critical role of TGFβ in disease development in MF, luspatercept and sotatercept may be effective in patients with MF [69,70,71]. Sotatercept has been evaluated in a single-center, investigator-initiated, phase 2 study (NCT01712308) as a single agent and in combination with ruxolitinib. Results from 13 evaluable patients receiving the combination of sotatercept and ruxolitinib were presented at the European Hematology Association (EHA) Annual Meeting in 2019. Three (23%) patients achieved an anemia response, defined as 1.5 g/dL increase in hemoglobin for ≥12 weeks or conversion from transfusion dependence to transfusion independence [34]. Luspatercept is being evaluated in a multicenter, phase 2 study of MF patients as a single-agent and in combination with ruxolitinib (NCT03194542). Preliminary results presented at the 2019 ASH Annual Meeting suggested encouraging response rates in transfusion-dependent patients receiving concurrent ruxolitinib. In this group, 6/19 (32%) patients achieved transfusion independence, leading to further expansion of this cohort. Despite low response rates in other cohorts (ruxolitinib-naïve transfusion-dependent/independent and transfusion-independent patients on ruxolitinib), which ranged from 10% to 21%, there is concern that the efficacy was underappreciated due to strict response criteria. To that end, 53% of transfusion-dependent patients receiving the combination experienced 50% reduction in RBC transfusion burden, and 57% of transfusion-independent patients receiving the combination achieved a mean hemoglobin increase of ≥1.5 g/dL. Treatment-related adverse events (AEs) occurring in more than 3% of patients included hypertension, bone pain, and diarrhea [33].

## 4. Combination Approaches in Accelerated- or Blast-Phase MPN

Transformation of MF to acute myeloid leukemia (AML) is associated with an extremely poor prognosis [72,73]. As a single agent, ruxolitinib has limited activity in this setting [74]. Low-intensity therapy with DNA methyltransferase inhibitors, such as azacitidine and decitabine, is preferred due to the poor outcomes seen with intensive chemotherapy [75,76]. In vitro studies have demonstrated sensitivity of post-MPN AML cells to decitabine, and synergism between ruxolitinib and decitabine in this setting has been hypothesized [77]. Clinically, the combination of decitabine and ruxolitinib has been assessed in two separate studies. Rampal et al. performed a phase 1 study of 21 patients with accelerated- or blast-phase MPN with escalating doses of ruxolitinib combined with standard decitabine dosing. Median OS was 7.9 months, and overall response rate was 43%. Ruxolitinib dosing did not appear to influence response or survival [36]. Bose et al. also assessed this combination in post-MPN AML patients, focusing on a ruxolitinib dose of 50 mg twice daily. Among 18 post-MPN AML patients treated at the recommended phase 2 dose of ruxolitinib, the overall response rate was 45%. Four patients were able to undergo allogeneic hematopoietic cell transplant. Median survival was 8.4 months [35]. The combination of ruxolitinib with low-intensity and high-intensity chemotherapy in blast-phase MPN patients has been limited to case series [78,79,80,81]. The addition of hydroxyurea to ruxolitinib is well tolerated and can provide cytoreductive benefits but is essentially palliative [82,83].

## 5. Novel Combinations with Clinical Experience

More recently, combination approaches have moved beyond the off-label repurposing of agents with clinical profiles that complement the effects of ruxolitinib. Rigorous preclinical investigations have identified rational targets that hold potential for synergism/additivity with JAK inhibition and/or the ability to recapture a clinical response in patients who have relapsed or progressive disease while being treated with a JAK inhibitor. In the following sections, we will highlight these combination approaches, combining the preclinical rationale with the clinical results in cases where these have been made public.

### 5.1. Phosphatidylinositol 3-Kinases/Protein Kinase B/Mammalian Target of Rapamycin (PI3K/AKT/mTOR) Inhibition (Umbralisib, Buparlisib, Parsaclisib)

One of the major pathways through which JAK2 mediates its downstream signaling is the PI3K pathway, which is activated in MPN [84]. Since AKT and mTOR are downstream targets of the PI3K pathway, they represent potential therapeutic vulnerabilities in MPN. The combination of ruxolitinib with either an AKT inhibitor (MK-2206) or an mTOR inhibitor (rad001) synergistically suppressed the survival of MPN model cells and inhibited the neoplastic growth of primary MPN patient cells [85,86]. Chronic treatment of MPN cells with ruxolitinib leads to incomplete inhibition of both JAK2 and its downstream target STAT5. Notably, it has been shown that combining PI3K and mTOR inhibitors with ruxolitinib could achieve maximal STAT5 inhibition in preclinical models of MPN [87]. A dual PI3K/mTOR inhibitor, BEZ235 when combined with ruxolitinib was more efficacious in suppressing proliferation and inducing apoptosis in MPN model cells and primary CD34^+^ MF cells than either of the inhibitors alone [88,89]. The cotreatment was also effective in reducing splenomegaly and improving survival of JAK2-V617F knock-in mice [89]. Interestingly, BEZ235 was lethal to MPN cells persistently surviving JAK2 inhibition, suggesting that PI3K/mTOR inhibition may overcome or prevent JAK2 inhibitor resistance [88].

The combination of umbralisib, a PI3K-delta inhibitor, and ruxolitinib was evaluated in a phase I study of MF patients who had predominantly been on ruxolitinib with a suboptimal or lost response (NCT02493530). Among 23 patients evaluable for response, two (9%) achieved CR and 11 (48%) met criteria for clinical improvement. Grade 3 diarrhea, neutropenia, and pancreatic enzyme elevation were seen in two patients each. Three patients were removed from the study due to AEs [40]. Buparlisib, a pan-PI3K, was combined with ruxolitinib in the phase 1b HARMONY study. This two-arm study enrolled JAK inhibitor-naïve and prior JAK inhibitor-treated patients. After six and 12 cycles, ≥50% reduction in spleen length was reported in 12/16 (75%) and 13/15 (87%) patients, respectively, in the JAK inhibitor-naïve arm. In patients who had received prior JAK inhibitor, 6/17 (35%) and 4/11 (36%) experienced ≥50% reduction in spleen length after six and 12 cycles, respectively. Enthusiasm regarding these results was tempered somewhat by the fact that only 5/9 (56%) and 3/7 (43%) of JAK inhibitor-naïve and prior JAK inhibitor-treated arms experienced ≥35% reduction in spleen volume after six cycles. Hematologic and nonhematologic AEs were common, with gastrointestinal, psychiatric, and infectious complications being notable [41].

More recently, parsaclisib, a potent and highly selective next-generation PI3K-delta inhibitor, has been added to ruxolitinib in patients with suboptimal response (NCT02718300). Patients were randomized to two different parsaclisib dosing schedules, with the more intensive (daily) dosing schedule appearing to be more effective. Interim results were presented at the 2020 EHA Annual Meeting. The primary endpoint was change in spleen volume at 12 weeks. At 12 weeks, 1/17 (6%) and 5/16 (31%) patients achieved a spleen response and symptom response, respectively. Interruption in parsaclisib was required in 25 of 51 patients. Notably, no pneumonitis, colitis, or dose-limiting diarrhea was observed [42].

### 5.2. Histone Deacetylase (HDAC) Inhibition (Pracinostat, Panobinostat)

HDACs, which comprise a family of 11 proteins, regulate the activity of both histone and nonhistone proteins by removing acetylation. Acetylation of histones supports transcriptional activation via relaxation of chromatin structure as well as by providing binding sites for recruitment of bromodomain co-transcriptional activating proteins [90]. Thus, while HDACs antagonize acetylation and inhibit gene expression, HDAC inhibitors enhance gene expression. The large number of HDACs, the nonspecificity of HDAC inhibitors, and the plethora of protein targets of HDACs not unexpectedly contribute to the pleiotropic effects of HDAC inhibitors [90]. Nonetheless, HDAC inhibitors (e.g., vorinostat, panobinostat, etc.) have been assessed preclinically and clinically for a variety of cancers, and some have been approved for various hematologic malignancies [91,92]. In preclinical MPN models, pan-HDAC inhibitors blocked growth and induced apoptosis in MPN model cells and primary cells from MPN patients [93,94,95,96]. Interestingly, HDAC inhibitors enhance the acetylation of HSP90, a stabilizing chaperone of JAK2, decreasing the affinity of HSP90 for JAK2; the treatment thus destabilizes the JAK2 protein [97]. As such, preclinical MPN studies have shown that HDAC inhibition leads to decreased JAK2 expression and signaling and sensitizes cells to JAK2 inhibition [93,98]. Notably, cells with mutationally active JAK2 were more sensitive to HDAC inhibition, which also displayed efficacy in murine models of MPN as a monotherapy and elicited augmented therapeutic responses in combination with ruxolitinib [93,95,98]. A recent study demonstrated that genetic deletion of HDAC11 expression impeded disease in a murine MPN model but did not affect normal hematopoiesis [99]. The development of more selective HDAC inhibitors will likely play an important role in determining the extent to which targeting this class of proteins can provide therapeutic efficacy with limited adverse effects.

Pracinostat is a pan-HDAC inhibitor that was assessed in combination with ruxolitinib in a phase 2 trial (NCT02267278). Patients received a lead-in of 12 weeks of ruxolitinib prior to the initiation of pracinostat. The combination proved challenging with 16 of 20 (80%) patients requiring dose interruption and a median time on pracinostat of only 5.3 months. Most patients discontinued due to hematologic toxicity, predominantly anemia and increasing transfusion requirements. Among 16 (80%) patients who achieved objective responses, 14 (88%) had their earliest response prior to starting pracinostat, suggesting that ruxolitinib was primarily driving the responses [43]. The combination of panobinostat and ruxolitinib has been studied in phase 1b studies in the United States and Europe (NCT01433445 and NCT01693601). Initial results of the European study, presented in 2014, appeared promising, with ≥50% reduction in palpable spleen length in 76% of patients and an acceptable toxicity profile [45]; however, the final results of this trial have not been published. Results from the American study were less encouraging, with frequent gastrointestinal toxicity and response rates similar to what would be expected with single-agent therapy. As a result, the phase 2 portion of the trial was not pursued. Still, seven (47%) patients experienced at least one-grade improvement in bone marrow fibrosis, though little change was noted in *JAK2* V617F mutant allele fraction [44].

### 5.3. Hedgehog Pathway Inhibitors (Sonidegib, Vismodegib)

The hedgehog signaling pathway is involved in various stages of hematopoiesis, from primitive hematopoiesis to maintenance of hematopoietic stem cell (HSC) populations [100]. Increased expression of hedgehog signaling pathway target genes has been demonstrated in primary samples from MPN patients and in a murine MF transplant model, suggesting that elevated hedgehog signaling is evident in MPN cells and thus may contribute to the neoplastic phenotypes of MPN. Combining a hedgehog pathway inhibitor, sonidegib, with JAK2 inhibition in a murine MF transplant model resulted in reduction of leukocytes, platelets, mutant allele burden, and bone marrow fibrosis, providing evidence that this pathway may be a therapeutic target for MPN [101].

Clinically, the hedgehog pathway inhibitors sonidegib (NCT01787552) and vismodegib (NCT02593760) have been studied in combination with ruxolitinib. Both studies enrolled MF patients naïve to JAK inhibition and demonstrated spleen and symptom responses comparable to what would be expected with single-agent ruxolitinib. Impacts on mutant allele burden and bone marrow fibrosis were modest. Tolerability was a concern, with 72% of patients treated with sonidegib and ruxolitinib experiencing AEs that required dose adjustment or interruption. The most common AEs occurring in at least 10% of patients were increased blood creatine phosphokinase, thrombocytopenia, anemia, muscle spasms, myalgia, and alopecia [48,49].

### 5.4. DNA Methyltransferase Inhibition (Azacitidine)

In contrast to the combination of decitabine and ruxolitinib, which has been assessed in patients with accelerated- and blast-phase MPNs, the combination of azacitidine and ruxolitinib has demonstrated encouraging results in patients with chronic-phase MF. Forty-six JAK inhibitor-naïve patients were enrolled to a phase 2 study that added azacitidine after a three-month run-in phase with single-agent ruxolitinib. Thirty-three of the 46 (72%) patients achieved an objective response while on study at a median of 1.8 months. Most responses were due to spleen and/or symptom responses and occurred before the addition of azacitidine. Spleen responses were assessed by palpation and were present in 21/34 (62%) patients at week 24 of the study. Among five transfusion-dependent patients, one (20%) achieved transfusion independence and 3/41 (7%) patients achieved ≥2 g/dL increase in their hemoglobin relative to baseline. Encouragingly, 3/15 (20%) patients who had abnormal cytogenetics achieved complete cytogenetic remission. Additionally, 16/24 (67%) had a decrease in JAK2 mutant allele burden at 24 weeks, and 19/31 (61.2%) patients experienced improvement in fibrosis during the study. The combination was relatively well tolerated with a median treatment duration of 18 months and rare grade 3/4 nonhematologic AEs [37].

### 5.5. Bromodomain and Extraterminal Protein (BET) Inhibition (CPI-0610)

The bromodomain family of proteins contain BET domains that bind to acetylated histones and regulate transcriptional activation of many genes that encode proteins that contribute to most aspects of cell biology and cancer [102]. BET inhibitors are small molecules that prevent BET domains from interacting with histones [103]. JQ1 was the first BET inhibitor introduced as a novel therapeutic to inhibit c-Myc, which plays important roles in oncogenesis but is not considered directly druggable [104,105]. BET inhibitors have rapidly progressed to clinical testing for a variety of cancers [106]. Preclinical MPN studies have shown that BET inhibition inhibits MPN cell growth and induces apoptosis of cell lines and primary cells from MPN patients [107,108,109]. These inhibitors display synergistic responses with JAK2 inhibition and are also effective at overcoming and preventing JAK2 inhibitor resistance. Proteins whose transcriptional control is disabled upon BET inhibition in MPN model cells include, in addition to c-Myc, PIM1 and BCL-xL (notably, two therapeutic targets described in this review), among others [107,108,110]. Importantly, BET inhibition has been shown to antagonize the disease phenotype in MPN mouse models and to augment disease inhibition by ruxolitinib, including impressive effects on allele burden and reversal of bone marrow fibrosis, in part by inhibiting the transcriptional activation of NFκB target genes [110]. Interestingly, BET inhibitory activity has been identified for the JAK2 inhibitor fedratinib [111,112]. The preclinical development of dual JAK2/BET inhibitors is ongoing with the goal of developing a single drug that can therapeutically target these critical proteins in MPN, potentially providing improved therapeutic efficacy with lower rates of drug resistance [113].

CPI-0610 is a novel BET inhibitor that is being evaluated in an ongoing multicohort study of MF patients, including assessing the combination with ruxolitinib (NCT02158858). In one arm, CPI-0610 is added to ruxolitinib in patients who have experienced suboptimal disease control. In another arm, the combination of CPI-0610 and ruxolitinib is being initiated in anemic MF patients who have never been treated with a JAK inhibitor. Interim results were most recently presented at the 2020 EHA Annual Meeting. In the “add-on” population, patients were stratified by transfusion status. Among evaluable transfusion-dependent patients, 11/32 (34.4%) converted to transfusion independence, 5/24 (20.8%) achieved spleen response, and 12/26 (46.2%) achieved a symptom response at week 24. At least one-grade improvement in bone marrow fibrosis was reported in 9/14 (64%) of patients. In the non-transfusion-dependent cohort, 4/18 (22.4%) achieved spleen response and 7/19 (36.8%) achieved symptom response at 24 weeks. Two of 12 patients (17%) demonstrated improvement in bone marrow fibrosis. Adverse events were typically related to thrombocytopenia, gastrointestinal symptoms, or infections [38].

The up-front combination of ruxolitinib and CPI-0610 led to spleen responses in 19/30 (63.3%) patients and symptom responses in 17/29 (58.6%) patients at 24 weeks. At least one-grade improvement in BM fibrosis occurred in 5/11 (46%) patients. Anemia (23%) and thrombocytopenia (20%) were relatively common; however, the incidence of treatment-emergent anemia was notably less that what had been reported with single-agent ruxolitinib. Most nonhematologic AEs were related to gastrointestinal symptoms or infections [114].

While still early, there is considerable optimism surrounding this combination. The ability to improve spleen volume, disease-related symptoms, and anemia while favorably impacting marrow fibrosis suggests a potential disease-modifying mechanism. Based on these encouraging results, a phase 3 study comparing up-front ruxolitinib + CPI-0610 to ruxolitinib + placebo is being planned.

### 5.6. B-Cell Lymphoma 2/B-Cell Lymphoma-Extra Large (Bcl-2/Bcl-xL) Inhibition (Navitoclax)

Members of the antiapoptotic Bcl-2 family of proteins (e.g., Bcl-2, Bcl-xL, Mcl-1, etc.) suppress apoptotic cell death by preventing proapoptotic caspase protease activation. Such antiapoptotic proteins are known transcriptional targets of JAK2/STAT signaling, suggesting that aberrant JAK2 activation may drive a strong antiapoptotic response in cells [115,116,117,118]. Numerous studies have demonstrated that combining inhibition of Bcl-2 family members with JAK2 inhibition significantly enhances the induction of apoptosis [116,119,120]. In addition, inhibition of Bcl-xL induced significant apoptotic cell death in JAK2 inhibitor-resistant cells driven by JAK2-V617F [119]. Interestingly, it has been shown that cell signaling mechanisms that promote the inactivation of BAD (BCL-2-associated death promoter), which functions to antagonize the antiapoptotic effects of Bcl-2 family members, are involved in mechanisms of JAK2 inhibitor resistance [121]. These data suggest inhibition of Bcl-2 family members, most notably Bcl-xL, may enhance anti-JAK2 therapies for MPN, potentially leading to improved initial responses as well as decreased development of JAK2 inhibitor resistance. Of note, Bcl-xL has been shown to be overexpressed in megakaryocytes from MPN patients compared to controls [122], and megakaryocytes are known drivers of myelofibrosis [123,124,125].

Navitoclax, a Bcl-2/Bcl-xL inhibitor, is being studied in an ongoing phase 2 study of MF patients with a suboptimal response to ruxolitinib (NCT03222609). The most recent update was presented at the EHA Annual Meeting in 2020. Among 30 patients evaluable for efficacy, a spleen response was observed in nine (30%) patients at week 24. Six (35%) patients experienced a symptom response. Eight of 32 (25%) patients had at least one-grade reduction in bone marrow fibrosis. Expectedly, thrombocytopenia (85%) and diarrhea (68%) were common. Further expansion of this trial has been planned [39].

### 5.7. Interferon Alpha

While the history of the use of interferon-α (IFN-α) in MPN patients goes back decades, its use has been limited by toxicity. Nonetheless, IFN-α has produced promising results in MPN patients, with complete remissions observed [126]. Toxicity has been mitigated by the use of pegylated (PEG) forms of IFN-α, allowing the potential for a greater number of patients to benefit from IFN-α therapy [126]. Significantly, treatment of JAK2-V617F mice with PEG-IFN-α preferentially depleted disease-initiating mutant JAK2-V617F long-term hematopoietic stem cells by inducing cell cycle activation and inducing differentiation, suggesting that IFN-α may antagonize clonal dominance of JAK2-V617F-expressing cells [127,128,129]. This provides a potential mechanism for the remissions observed in IFN-α-treated patients. Ruxolitinib does not readily reduce the malignant clone in patients and has no effect on disease-driving stem cells in MPN model mice [129,130]. INF-α is a rare example of an MPN therapy that can induce remission in patients and target disease-driving stem cells in murine MPN models. Importantly, while IFN-α signals through JAK1, cotreatment with ruxolitinib did not antagonize the effects of IFN-α in disease-driving stem cells, suggesting that the combination of IFN-α and JAK2 inhibition may hold promise to not only provide rapid quality of life improvements through the anti-inflammatory effects of JAK2 inhibition but also provide significant long-term disease-modifying and remission-inducing potential for patients [129].

The combination of ruxolitinib and low-dose PEG-IFNα2 has been clinically assessed in MPN patients. A phase 2 study that enrolled PV and MF patients (EudraCT2013-003295-12) assessed low-risk MF patients, many of whom had previously been treated with interferon. Fifteen out of 18 (83%) patients had low or int-1 risk by DIPSS plus, making this trial unique in comparison to most MF trials, which enroll a disproportionate amount of higher-risk patients. By in large, patients had preserved blood counts (median hemoglobin 12.8 g/dL) with rare splenomegaly (17%) and mild to moderate symptom burden (median MPN SAF TSS 24). Among 18 MF patients, 10 (56%) responded, suggesting activity of the combination in this lower-risk cohort [46]. An ongoing phase 1/2 study assessing the combination of PEG-IFNα2 and ruxolitinib (NCT02742324) is in progress and aims to identify the optimal combination dose and assess efficacy in MF patients who have not been treated with either agent prior to the study. Early results presented at the 2018 ASH Annual Meeting suggested the combination was safe, with no dose-limiting toxicities seen with doses up to PEG-IFNα2 at 135 mcg/week combined with ruxolitinib 20 mg BID. In the phase 2 portion, patients will be randomized between this dose and PEG-IFNα2 at 135 mcg/week and ruxolitinib 15 mg BID. Preliminary efficacy data showed encouraging results in terms of spleen size reduction and reduction of allele burden of mutant clone and other beneficial impacts on immune cells. Further results from the phase 2 portion of this study are eagerly anticipated [47].

### 5.8. Antifibrotic Agents

Since JAK2 inhibition alone does not significantly improve bone marrow fibrosis in MF, agents that target fibrosis are also being tested as combination partners for ruxolitinib in MF. Lysil-oxidase-like-2 (LOXL2) is a monoamine oxidase that promotes collagen fiber formation in the extracellular matrix [131]. LOXL2 has been reported to be overexpressed in patients with PMF, suggesting it may play a role in bone marrow fibrosis [132]. LOXL2 expression in megakaryocytes may contribute to bone marrow fibrosis, which can be inhibited by inhibition of LOXL2 [133]. MF patients contain neoplastic fibronectin and collagen-producing fibrocytes, which are inhibited by serum amyloid P (pentraxin-2). A recombinant form of human pentraxin 2, PRM-151, inhibited bone marrow fibrosis in a murine model of MF [134]. The glycogen synthase kinase-beta (GSK-3β) inhibitor 9-ING-41 blocked pulmonary fibrosis induced by TGF-β, suggesting an analogous potential in bone marrow fibrosis [135].

Simtuzumab is a monoclonal antibody inhibitor of LOXL2. Due to its potential role as an antifibrotic agent, it was assessed in a phase 2 study of MF patients (NCT01369498). Within this study, 30 patients received a combination of simtuzumab and ruxolitinib. Three patients (10%) experienced a reduction in bone marrow fibrosis, while six (20%) patients experienced worsening fibrosis. One patient (5%) had a hemoglobin improvement and 0/12 (0%) experienced a spleen response. While grade ≥3 AEs were rare, gastrointestinal toxicity (diarrhea, nausea, constipation), fatigue, anemia, and epistaxis were observed in ≥30% of patients treated with the combination [50].

PRM-151 is being studied in MF as a single agent and in combination with ruxolitinib (NCT01981850). Preliminary data presented at the 2018 ASH Annual Meeting analyzed nine patients who experienced clinical benefit from the combination and entered into the open-label extension portion of the study. Favorable impacts on spleen size and disease-related symptoms and improvement in bone marrow fibrosis were noted in combination with acceptable tolerability. Despite encouraging preliminary results, the future development of PRM-151 in MF is uncertain [136].

On the heels of simtuzumab and PRM-151, the GSK-3β inhibitor 9-ING-41 is entering into a phase 2 clinical trial (NCT04218071) in MF patients that allows for combination with ruxolitinib.

## 6. Novel Combinations Pending Clinical Experience

### 6.1. Aurora Kinase Inhibition (Alisertib)

Aberrant regulation of megakaryocytes plays a significant driving role on the MPN phenotype [123,124,125]. Inhibition of aurora kinase A induces the differentiation and apoptosis of megakaryocytes [137]. Aurora kinase A was found to be upregulated in MF patients, and the aurora kinase A inhibitor MLN8237 induced differentiation and inhibited cell growth of CD34^+^ cells from MF patients [138]. MLN8237 displayed impressive efficacy against disease phenotypes in murine MPN models. Importantly, deletion of one allele that encodes aurora kinase A completely abrogated myelofibrosis in a murine MPN model, providing clear genetic evidence that decreasing aurora kinase A activity may provide therapeutic benefit in MF. In addition, MLN8237 synergized with ruxolitinib to antagonize disease, including elimination of myelofibrosis [138].

Alisertib (MLN8237) was assessed in a phase 1 trial of 24 MF patients. Results from this study indicated that alisertib was well tolerated and displayed efficacy at decreasing spleen size and symptom burden in about 30% of the patients. Importantly, alisertib reduced megakaryocytes and reduced myelofibrosis in 5/7 patients evaluable for sequential assessment. These results strongly support additional evaluation of alisertib as an MPN therapeutic, including in combination with JAK2 inhibition in MF patients [139].

### 6.2. Heat Shock Protein 90 (HSP90) Inhibition (PU-H71)

JAK2 overexpression has been observed as being associated with chronic JAK2 inhibition, suggesting that JAK2 inhibitors, such as ruxolitinib, may lead to an increase in their therapeutic target, which may contribute to drug resistance [140]. HSP90 is a molecular chaperone that ensures proper folding and stability of several oncoproteins, including JAK2, and hence serves as an attractive therapeutic target [141,142]. Inhibition of HSP90 in MPN cells increased degradation of JAK2, resulting in inhibition of JAK2-mediated signaling [143,144]. Combination of HSP90 and JAK2 inhibition synergistically suppressed the neoplastic growth of primary MPN patient cells and could overcome JAK2 inhibitor resistance [144]. The HSP90 inhibitor PU-H71 has been shown to be effective in mouse models of MPN and against primary cells from patients [145] and enhanced the efficacy of JAK2 inhibition in therapeutic models of MPN [146]. Addiction of mutant JAK2-V617F MPN cells to HSP90 might be responsible for their greater sensitivity to HSP90 inhibition compared to cells expressing only wild-type JAK2, which may be important with respect to a therapeutic window in patients. Two phase 1b clinical trials (NCT03935555, NCT03373877) to test the safety, tolerability, pharmacokinetics, and efficacy of PU-H71 are being carried out in primary and secondary MF patients who have persistent or worsening disease while undergoing ruxolitinib therapy.

### 6.3. Lysine-Specific Demethylase 1 (LSD1) (Bomedemstat)

LSD1 regulates epigenetic control of gene expression by demethylating lysine residues on histone H3 [147]. LSD1 plays critical roles in hematopoiesis, including maturation of megakaryocytes, which play a driving function in MPN [123,124,125,148,149]. LSD1 inhibition has been shown to target the function of myeloid leukemic stem cells [150]. It is reportedly overexpressed in half of MPN patients, including 58% of MF patients assessed [151]. In mouse models of MPN, the LSD1 inhibitor IMG-7289 improved all aspects of disease, including mature cell counts, allele burden, cytokine levels, bone marrow fibrosis, and increased survival of mice. Low-dose combinations of IMG-7289 and ruxolitinib led to further improvement of disease parameters compared to monotherapies, supporting the clinical assessment of this combination in MF [152].

As a single agent, bomedemstat (IMG-7289) has demonstrated improvement in spleen volume and disease-related symptoms in MF patients intolerant or resistant to ruxolitinib [153]. Future combination studies are planned.

### 6.4. Mouse Double Minute 2 Homolog (MDM2) Inhibition (Idasanutlin, KRT-232)

JAK2-V617F has been shown to destabilize the p53 tumor suppressor by enhancing expression of MDM2, a ubiquitin ligase that controls p53 activity, and inhibition of MDM2 in these cells enhances apoptotic responses to stress [154]. MDM2 inhibition has also been shown to enhance apoptosis of polycythemia vera (PV) progenitor cells [155]. Furthermore, the balance of stress responses in MPN progenitors is believed to be tightly regulated to survive the stress of an inflammatory microenvironment, enhancing the survival of disease-driving cells [156]. In addition, the loss of p53 function promotes post-MPN leukemic transformation [77,157]. These data suggest that control of p53 function may play key roles in disease progression.

In an early-phase study of PV patients, the MDM2 inhibitor idasanutlin demonstrated clinical efficacy and encouraging reductions in JAK2 mutant allele burden [158]. The MDM2 inhibitor KRT-232 is currently being evaluated in PV and MF patients (NCT03669965 and NCT03662126), with preliminary results showing clinical activity in higher-risk MF patients who experienced relapse or refractory disease during treatment with a JAK inhibitor [159]. Siremadlin (HDM 201) is an MDM2 inhibitor being evaluated in combination with ruxolitinib as part of a platform study of novel combinations that also includes a P-selectin inhibitor (crizanlizumab) and a TIM-3 inhibitor (MBG453) (NCT04097821).

### 6.5. Nuclear Factor Kappa-Light-Chain-Enhancer of Activated B Cell (NFκB) Inhibition (Pevonedistat)

Elevated inflammatory cytokines and cytokine signaling play a significant role in the pathogenesis of MPNs and are associated with poor outcomes [160]. This contribution of cytokine activation is recapitulated in murine models of MPN, where the NF*κ*B pathway has been identified as playing a role in the upregulation of cytokine expression [110]. Likewise, NF*κ*B activation was identified in CD34^+^ cells from MF patients, and inhibition of NF*κ*B blocked proliferation of these cells [161]. JAK2 inhibitors can reduce the expression of certain cytokines in patients, which likely contributes to the improved quality of life afforded by anti-JAK2 therapies [162]. However, not all cytokines are responsive to JAK2 inhibition, suggesting other pathways, such as NF*κ*B, may be more critical in regulating cytokine expression [163]. Cytokine signaling, including TNFα, which activates NF*κ*B, may also contribute to the neoplastic growth of MPN-driving stem and progenitor cells [164]. Thus, antagonizing cytokine/NF*κ*B signaling could provide a valuable therapeutic approach for MPNs. One approach to target NF*κ*B is to utilize pevonedistat (MLN4924), a neddylation inhibitor that blocks the degradation of the NF*κ*B inhibitor I*κ*B, thus preventing NF*κ*B-mediated activation of its target genes [165,166,167]. Preclinical studies have demonstrated that pevonedistat indeed blocks induction of inflammatory cytokines from an MPN model cell line as well as primary MF monocytes [163]. A clinical trial (NCT03386214) to assess the activity of pevonedistat and ruxolitinib in MF is ongoing.

### 6.6. Protein Arginine Methyltransferase 5 (PRMT5) Inhibition

PRMT5 is a protein arginine methyltransferase that catalyzes the methylation of both histone and nonhistone proteins, contributing to epigenetic regulation of gene expression and function of proteins regulating a variety of cellular pathways [168,169]. PRMT5 was found overexpressed in a subset of MPN patients, and inhibition of its activity altered expression of many genes deregulated in MPN, suggesting that PRMT5 may play a role in MPN pathogenesis [170]. In fact, the PRMT5 inhibitor C220 displayed efficacy against the disease phenotypes of MPN mouse models, including inhibition of bone marrow fibrosis, and for some parameters of disease, it produced enhanced effects with ruxolitinib in these models. Mechanistic studies revealed PRMT5 inhibition altered the methylation and activity of the E2F transcription factor, associating with increased cell cycle arrest and apoptosis [170]. PRMT5 inhibitors are being assessed in clinical trials for various indications, including PRT543, which is being tested in advanced solid and hematologic cancers, including MF patients (NCT03886831). These preclinical MPN studies provide strong support for testing the combination of PRMT5 and JAK2 inhibition in MPN [170].

### 6.7. Proviral Integration Site for Moloney Murine Leukemia Virus (PIM) Inhibition (INCB053914)

PIM kinases signal, in part, to inhibit apoptosis, activate the mTOR pathway, and regulate MYC function, and the genes that encode PIMs are proto-oncogenes, as aberrant expression of PIMs induce neoplastic growth [171]. PIMs thus provide a potential target for cancer therapeutics. However, they are direct transcriptional targets of STAT proteins, suggesting that targeting PIMs may be of particular value in neoplasms induced by activated JAK/STAT signaling [171]. Preclinical studies have shown that inhibition of PIM kinases in MPN model cells leads to reduced MYC protein levels and inhibition of mTOR signaling and cell proliferation. Combining PIM inhibition with JAK2 inhibition synergistically induced apoptosis and sensitized MPN cells to JAK2 inhibition, including synergistically suppressing the neoplastic growth of primary cells from MPN patients [172,173,174]. Interestingly, the PIM kinase inhibitor INCB053914 antagonized the development of ruxolitinib-persistent disease progression in a mouse model of MPN [174]. PIM inhibitors may therefore enhance the efficacy of JAK2 inhibition in MPN patients. Within the context of a broadly enrolling phase 1/2 clinical trial, the combination of INCB053914 and ruxolitinib is being specifically assessed in MF patients with a suboptimal response to ruxolitinib (NCT02587598).

### 6.8. Other Combinations

Other combinations that are supported by preclinical data and have yet to reach, or just recently entered, clinical testing are worth noting. Recent reports have suggested that disease progression in MF patients treated with ruxolitinib is associated with activation of receptor tyrosine kinase and RAS pathway mutations, and RAS pathway mutations, including *PTPN11* (which encodes the SHP2 phosphatase), confer poor prognosis [24,25]. Notably, inhibition of MEK, a mediator of MAP kinase signaling, has been shown to be effective in murine models as a single agent as well as in combination with ruxolitinib [175]. Similarly, SHP2 mediates activation of the RAS signaling pathway, and targeting SHP2 can synergize with ruxolitinib in preclinical models and may antagonize disease in a murine model of MPN [176]. JAK2 inhibitor resistance may be mediated in part by JAK1 activation, which also functions as a critical mediator of inflammatory cytokine signaling, suggesting that JAK1 inhibition may be effective in MPN and/or enhance the efficacy of JAK2 inhibition [140,177]. The JAK1 inhibitor itacitinib (INCB039110) is being assessed alone and in combination with ruxolitinib in MF (NCT03144687). Preliminary results suggest that single-agent itacitinib demonstrates limited hematologic toxicity and meaningful symptom relief but only modest reductions in spleen size (NCT01633372) [178]. Telomerase is a ribonuclear protein complex that extends the length of telomeres and is preferentially activated in cancer cells relative to mature somatic cells. Imetelstat is a telomerase inhibitor that has shown the ability to induce morphologic and molecular remissions in a subset of MF patients [179]. This appears to be due to a preferential depletion of MF hematopoetic stem/progenitor cells (HSCs/HPCs) compared to normal HSCs/HPCs [180]. Sequential combination of ruxolitinib and imetelstat may potentiate this effect [181]. While imetelstat continues to be developed as a single agent in MF, there are no current trials investigating imetelstat in combination with a JAK inhibitor.

## 7. Discussion/Conclusions

The field of ruxolitinib combination therapy in MF is still in its relative infancy. As results emerge from several trials assessing combination approaches, interpreting these results will become a challenge. Discrepancies in study design, endpoints, and patient populations will impede fair comparisons. Ultimately, promising combination therapies will need to be assessed in randomized, phase 3 studies, but even then the appropriate endpoints are unclear and may differ by agent. The importance of this cannot be discounted given past challenges of drug development in MF.

Planning for phase 3 combination studies is already underway for several agents based on promising phase 2 results. While these trials will hopefully demonstrate improved disease control, it is important to note that response rates seen in phase 2 studies typically dwarf those seen in subsequent randomized controlled trials. Prior to demonstrating spleen volume response rates of 32–42% at 24 weeks in the COMFORT studies [3,4], ruxolitinib (INCB018424) led to spleen responses in 52% of patients at week 12 with the favored dosing regimen in a phase 2 study [162]. Similarly, phase 2 studies of momelotinib and fedratinib reported spleen volume responses ranging from 54% to 60% before exhibiting more modest responses in randomized studies [182,183].

Nevertheless, phase 3 trials promise to provide comparative data to support or refute the superiority of combination approaches over standard of care. Yet, in a disease where ≥35% reduction in spleen volume and 50% reduction in total symptom score represent the most common measures of drug efficacy, it is not clear whether randomized trials will adequately determine if a combination approach represents a therapeutic breakthrough. Unfortunately, there are no standardized markers of disease modification, and trials have been hesitant to utilize overall survival as an endpoint in such a prognostically heterogeneous disease. Progression-free survival holds potential as an endpoint, but the current definition only captures spleen growth and leukemic transformation, thus ignoring progressive disease-related symptoms or cytopenias [184].

Despite these challenges, the burgeoning field of ruxolitinib combination therapies promises to yield an improved understanding of a complex disease and will hopefully lead to therapeutic advances. Undoubtedly, patients with MF are in desperate need of therapeutic options and now, more than ever before, there is reason for considerable optimism.

## Figures and Tables

**Table 1 cancers-12-02278-t001:** Clinical trials assessing ruxolitinib in combination with another agent in patients with myelofibrosis or myelofibrosis-related conditions.

Agent	Mechanism/Target	Trial Identifier	Phase	Primary Efficacy Endpoint	Results
Danazol	Androgen	NCT01732445	2	Best overall response	Published (Gowin et al., 2017) [29]
Lenalidomide	IMiD	NCT01375140	2	Best overall response	Published (Daver et al., 2015) [30]
Pomalidomide	IMiD	NCT01644110	1/2	Anemia Response	Interim results at ASH 2019 [31]
Thalidomide	IMiD	NCT03069326	2	Best overall response	Interim results at ASH 2019 [32]
Luspatercept	TGFβ	NCT03194542	2	Anemia response	Interim results at ASH 2019 [33]
Sotatercept	TGFβ	NCT01712308	2	Anemia response	Interim results at EHA 2019 [34]
INCB000928	TGFβ	NCT04455841	1/2	N/A*	N/A
Decitabine	DNMT	NCT02257138	1/2	Best overall response	Published (Bose et al., 2020) [35]
		NCT02076191	1/2	N/A*	Phase 1 published (Rampal et al., 2018) Interim results of phase 2 at ASH 2018 [36]
		NCT04282187	2	Receipt of HCT	N/A
Azacitidine	DNMT	NCT01787487	2	Best overall response	Published (Masarova et al., 2018) [37]
Enasidenib	IDH2	NCT04281498	2	Best overall response	N/A
CPI-0610	BET	NCT02158858	1/2	Spleen response; Transfusion Independence	Interim results at EHA 2020 [38]
Navitoclax	Bcl-2/Bcl-xL	NCT03222609	2	Spleen response	Interim results at EHA 2020 [39]
APG-1252	Bcl-2/Bcl-xL	NCT04354727	1/2	Spleen or Symptom response	N/A
Umbralisib	PI3K-delta	NCT02493530	1	N/A*	Interim results at EHA 2018 [40]
Buparlisib	Pan-PI3K	NCT01730248	1	N/A*	Published (Durrant et al., 2019) [41]
Parsaclisib	PI3K-delta	NCT02718300	2	Spleen response	Interim results at EHA 2020 [42]
Pracinostat	Pan-HDAC	NCT02267278	2	Best overall response	Published (Bose et al., 2019) [43]
Panobinostat	Pan-HDAC	NCT01693601	1	N/A*	Published (Mascarenhas et al., 2020) [44]
		NCT01433445	1	N/A*	Interim results at ASCO 2014 [45]
PEG-IFNα2	Interferon	EudraCT 2013-003295-12	2	Best overall response	Published (Sorensen et al., 2020) [46]
		NCT02742324	1/2	Spleen response (by palpation)	Interim results at ASH 2018 [47]
Sonidegib	Hedgehog pathway	NCT01787552	1/2	Spleen response	Published (Gupta et al., 2020) [48]
Vismodegib	Hedgehog pathway	NCT02593760	1	N/A*	Published (Couban et al., 2018) [49]
Pevonedistat	NEDD8	NCT03386214	1	N/A*	N/A
INCB053914	PIM	NCT02587598	1	N/A*	N/A
PIM447, LEE011	PIM, CDK4/6	NCT02370706	1	N/A*	N/A
Simtuzumab	LOXL2	NCT01369498	2	Fibrosis improvement	Published (Verstovsek et al., 2017) [50]
9-ING-41	GSK-3β	NCT04218071	2	Best overall response	N/A
PRT543	PRMT5	NCT03886831	1	N/A*	N/A
Itacitinib	JAK1	NCT03144687	2	Change in spleen volume	N/A
Siremadlin Crizanlizumab MBG453	MDM2 P-selectin TIM-3	NCT04097821	1/2	Best overall response	N/A

IMiD: immunomodulatory imide agent; ASH: American Society of Hematology; TGFβ: transforming growth factor beta; EHA: European Hematology Association; DNMT: DNA methyltransferase; HCT: hematopoietic cell transplant; IDH2: isocitrate dehydrogenase-2; BET: bromodomain and extraterminal domain; Bcl-2/Bcl-xL: B-cell lymphoma-2 and B-cell lymphoma-extra large; PI3K: phosphatidylinositol-3-kinase; HDAC: histone deacetylase; PIM: proviral integration Moloney virus; CDK4/6: cyclin-dependent kinase 4/6; LOXL2: lysyl oxidase-like 2; GSK-3β: glycogen synthase kinase 3 beta; PRMT5: protein arginine N-methyltransferase 5; JAK1: Janus kinase-1; MDM2: mouse double minute 2 homolog (MDM2); TIM-3: T-cell immunoglobulin mucin-3; N/A*: no primary efficacy endpoint due to trial design.

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
