# Peer review of "Finding a Jill for JAK: Assessing Past, Present, and Future JAK Inhibitor Combination Approaches in Myelofibrosis"

_cancers, 2020, doi:10.3390/cancers12082278_

Round 1

Reviewer 1 Report

As a subpopulation of myeloproliferative neoplasms, myelofibrosis (MF) patients had severe clinical features and poor prognosis. The vast majority of MF patients harbor a phenotype-driving activating mutation in one of three genes: JAK2, MPL, and CALR, all of which lead to upregulation of the JAK/STAT pathway. Development of JAK2 inhibitors arises great clinical benefits for MF patients. However, it had proved that JAK2-inhibitors had several drawbacks, which trigger the exploration of combination strategies. Herein, Andrew T. Kuykendall et al summarized the progress on JAK2-inhibitor combination approaches in MF, which is necessary for future study and highlighted emerging strategies of particular interest. The review article was well organized and comprehensive. I do not have particular concerns or further comments. 

Author Response

Reviewer 1 – Response: There were no specific comments to address from Reviewer 1. We thank the reviewer for their positive review.

Reviewer 2 Report

This comprehensive, timely, and well-written review covers the interesting and important topic of various (potential) combination partners for JAK inhibitors in myelofibrosis.

However, there are several points that should be addressed:

Major points:

  • Page 2, lines 58-60: the statement that "Beyond an impact on spleen size and disease-related symptoms, ruxolitinib has shown favorable impact on nutritional parameters, hepatomegaly, and stabilization – but not resolution of JAK2 allele burden and bone marrow fibrosis" is not entirely valid anymore, given that a) individual patients have been described who have shown marked reductions in JAK2 allele burden down to negative resutls (Deininger et al Blood 2015 PMID: 26228487) and b) the degree of BM fibrosis was decreased in a substantial fraction of patients after long-term expsosure to RUX (Kvasnicka et al JHO 2018 PMID: 29544547). This should be incorporated into the manuscript.
  • Page 3, line 105: in general and also here, not only the efficacy data of the clinicla trials that are cited are important but also the tolerability. Please add tolerability data here (and for the other trials as well, where missing).
  • Page 10, line 446; page 11, line 501: the trials cited as references 148 and 167 should be described in more detail, similar to the other cited trials (presented at EHA, ASCO, or ASH)
  • What about imetelstat? The published trials (e.g. Tefferi et al NEJM 2015 etc) should be cited as well (and added to the table).

Minor points:

  • Page 2, line 86: the subtitle "Killing Two Birds with Two Stones – Efforts to Manage Anemia during Ruxolitinib Therapy" is unclear. Which are the two birds? If anemia is one, whcih one is the other one? Please specify in the subtitle.
  • Page. 9, line 387: in chapter 5, the concepts with fully published studies (e.g. hedgehog signaling inhibitor or buparlisib trials and others) should be listed before the more experimental trials with more preliminary results.
  • Page 9, line 403: starting from here, most trials reported subsequently in the manuscript report the use of single agents as opposed to combinations with a JAK inhibitor (with the prospect of being combined later on). Therefore, a new subtitle should be included to allude to this fact.

Author Response

Reviewer 2 – Response: We thank the reviewer for their comments and helpful suggestions. We have modified our submission to address these comments as indicated below.

Major Points

  • Reviewer 2: “Page 2, lines 58-60: the statement that "Beyond an impact on spleen size and disease-related symptoms, ruxolitinib has shown favorable impact on nutritional parameters, hepatomegaly, and stabilization – but not resolution of JAK2 allele burden and bone marrow fibrosis" is not entirely valid anymore, given that a) individual patients have been described who have shown marked reductions in JAK2 allele burden down to negative resutls (Deininger et al Blood 2015 PMID: 26228487) and b) the degree of BM fibrosis was decreased in a substantial fraction of patients after long-term expsosure to RUX (Kvasnicka et al JHO 2018 PMID: 29544547). This should be incorporated into the manuscript.”

Response: Deininger et al shows that 2.5% (6/236) of patients achieved a CMR which, while encouraging, suggests that this is a very rare event. Moreover, beyond the change in allele frequency seen in the first 6 months (between 10-15% reduction), the remaining change in allele fraction represents that seen in patients continuing to respond. Similarly, in the study by Kvasnicka, the ruxolitinib-treated cohort is heavily weighted towards patients who experienced an excellent response to ruxolitinib since it required bone marrow biopsies at baseline and 24 months. Those patients who lost response were thereby excluded. We agree that there may be an impact of ruxolitinib on variable allele fraction and/or bone marrow fibrosis, but this impact is, at best, modest compared to the effects of ruxolitinib on spleen size/disease related symptoms. We have edited our submission (and added the suggested references) in response to this comment.

Lines 59-62 now read: “Beyond an impact on spleen size and disease-related symptoms, ruxolitinib has shown favorable impact on nutritional parameters and hepatomegaly, while its impact on JAK2 allele burden and bone marrow fibrosis has been modest, with more significant potential impact emerging with long-term treatment [7,9-12].

  • Reviewer 2: “Page 3, line 105: in general and also here, not only the efficacy data of the clinicla trials that are cited are important but also the tolerability. Please add tolerability data here (and for the other trials as well, where missing).”

Response: Page 3, line 105: We agree that tolerability is as important as efficacy in combination studies given the high risk for overlapping toxicity profiles. While the focus of this review is on the preclinical rationale for clinical efficacy, we have attempted to include reference to tolerability in all combinations where it is available. Since much of the data is from interim analyses and, therefore, preliminary in nature – we have elected to provide summary statements regarding tolerability as opposed to specific percentages as this is likely to change in the final analyses. In response to this comment, we have edited our submission to reference tolerability as possible, in Lines 109, 288, 303, and 504.

  • Reviewer 2: “Page 10, line 446; page 11, line 501: the trials cited as references 148 and 167 should be described in more detail, similar to the other cited trials (presented at EHA, ASCO, or ASH)”

Response: Page 10, line 446; page 11, line 501: In reference to page 10, line 446, the two trials referenced with idasanutlin and KRT-232 are given less weight due to the fact that the first is in polycythemia vera patients and the second is a single-agent; therefore, results from these trials are somewhat outside the context of our review which focuses on combination therapy for MF patients. We allude to these trials because combination strategies are actively being pursued and these single-agent trials provide clinical context for these combination efforts. In reference to page 11, line 501, we agree that discussion of trials using simtuzumab and PRM-151 should have been given more weight. This section has been updated to discuss the available clinical data using these agents in combination with ruxolitinib.

Lines 492 – 506 now read: “Simtuzumab is a monoclonal antibody inhibitor of LOXL2. Due to its potential role as an anti-fibrotic agent, it was assessed in a phase 2 study of MF patients (NCT01369498). Within this study, 30 patients received a combination of simtuzumab and ruxolitinib. Three patients (10%) experienced a reduction in bone marrow fibrosis while 6 (20%) patients experienced worsening fibrosis. One patient (5%) had a hemoglobin improvement and 0/12 (0%) experienced a spleen response. While grade ≥ 3 AEs were rare, gastrointestinal toxicity (diarrhea, nausea, constipation), fatigue, anemia, and epistaxis were observed in ≥ 30% of patients treated with the combination [135].

PRM-151 is being studied in MF as a single agent and in combination with ruxolitinib (NCT01981850). Preliminary data presented at the 2018 annual ASH meeting analyzed 9 patients who experienced clinical benefit from the combination and entered into the open label extension portion of the study. Favorable impacts on spleen size, disease-related symptoms, and improvement in bone marrow fibrosis were noted in combination with acceptable tolerability. Despite encouraging preliminary results, the future development of PRM-151 in MF is uncertain [136].

  • Reviewer 2: “What about imetelstat? The published trials (e.g. Tefferi et al NEJM 2015 etc) should be cited as well (and added to the table).”

Response: While certainly an agent with potential promise for refractory myelofibrosis patients, to our knowledge, imetelstat has not been clinically studied in combination with ruxolitinib and there are no known immediate plans for combination efforts. As this review focuses on combination therapies, a focused review on imetelstat is somewhat beyond the scope of this review. However, in response to this comment, in the section on “other combinations” we have referenced the existing preclinical rationale behind an imetelstat combination approach. While important, imetelstat trials were not added to the table since the table references combination trials.

The following text has been added to this submission (Lines 697-700): “Telomerase is a ribonuclear protein complex that extends the length of telomeres and is preferentially activated in cancer cells relative to mature somatic cells. Imetelstat is a telomerase inhibitor that has shown the ability to induce morphologic and molecular remissions in a subset of MF patients [180]. This appears to be due to a preferential depletion of MF hematopoetic stem/progenitor cells (HSCs/HPCs) compared to normal HSCs/HPCs) [181]. Sequential combination of ruxolitinib and imetelstat may potentiate this effect [182]. While imetelstat continues to be developed as a single-agent in MF, there are no current trials investigating imetelstat in combination with a JAK inhibitor.”

Minor Points

  • Reviewer 2: “Page 2, line 86: the subtitle "Killing Two Birds with Two Stones – Efforts to Manage Anemia during Ruxolitinib Therapy" is unclear. Which are the two birds? If anemia is one, whcih one is the other one? Please specify in the subtitle.”

Response: This has been clarified and hopefully this makes more sense. The subtitle now reads “3. Killing Two Birds with Two Stones – Efforts to Manage Anemia while Controlling Splenomegaly/Symptoms with Ruxolitinib Therapy

  • Reviewer 2: “ 9, line 387: in chapter 5, the concepts with fully published studies (e.g. hedgehog signaling inhibitor or buparlisib trials and others) should be listed before the more experimental trials with more preliminary results.”

Response: Thank you for this suggestion. This section has been reorganized to prioritize fully published studies (Hedgehog, HDAC, HMA, PI3K).

  • Reviewer 2: “Page 9, line 403: starting from here, most trials reported subsequently in the manuscript report the use of single agents as opposed to combinations with a JAK inhibitor (with the prospect of being combined later on). Therefore, a new subtitle should be included to allude to this fact.”

Response: We thank the reviewer for this helpful suggestion. We have created a new section separating combination trials with clinical results and those that are pending clinical results. To be clear, we tried to limit our discussion to combinations that have planned clinical trials with NCT numbers that can be referenced. The two subtitles now included are: “5. Novel Combinations with Clinical Experience” and “6. Novel Combinations Pending Clinical Experience”.

Reviewer 3 Report

The review is very well organized and gives a full overview of the different therapy associations available at the moment.

Only minor improvements are suggested:

1) Row 67: "Additionally, due to impacts on innate and adaptive immunity (ref?)...". Hera a reference is missing (e.g. Elli EM et al. Mechanisms Underlying the Anti-inflammatory and Immunosuppressive Activity of Ruxolitinib. Front Oncol. 2019;9:1186. Published 2019 Nov 7. doi:10.3389/fonc.2019.01186);

2) Rows 88-92: the study by Crisà E. et al. The use of erythropoiesis-stimulating agents is safe and effective in the management of anaemia in myelofibrosis patients treated with ruxolitinib. Br J Haematol. 2018;182(5):701-704. doi:10.1111/bjh.15450 should be reported;

3) While the review is very updated and also includes works presented at ASH and EHA meetings, the association of ruxolitinib and deferasirox presented at ASH 2019 (Elli E et al. Concomitant Treatment with Ruxolitinib and Deferasirox in the Management of Iron Overload in Patients with Myelofibrosis: A Multicenter Italian Experience. Blood (2019) 134 (Supplement_1): 839. should be reported. I am not sure whether it is more appropriate to include it in the "3. Killing Two Birds with Two Stones – Efforts to Manage Anemia during Ruxolitinib Therapy" (row 86) paragraph or in the "5.15. Other combinations" (row 503) paragraph. I would leave the choiche to the authors.

Author Response

Reviewer 3 – Response: We thank the reviewer for their positive review, and have addressed the following minor points made by the reviewer.

  • Reviewer 3: “1) Row 67: "Additionally, due to impacts on innate and adaptive immunity (ref?)...". Hera a reference is missing (e.g. Elli EM et al. Mechanisms Underlying the Anti-inflammatory and Immunosuppressive Activity of Ruxolitinib. Front Oncol. 2019;9:1186. Published 2019 Nov 7. doi:10.3389/fonc.2019.01186);”

Response: This reference has been added as suggested (Line 73, reference 19).

  • Reviewer 3: “2) Rows 88-92: the study by Crisà E. et al. The use of erythropoiesis-stimulating agents is safe and effective in the management of anaemia in myelofibrosis patients treated with ruxolitinib. Br J Haematol. 2018;182(5):701-704. doi:10.1111/bjh.15450 should be reported;”

Response: This reference was added (reference 30) with a brief allusion to this retrospective study.

The following text has been added in response to this comment (Lines 98-100): “A subsequent retrospective analysis suggested a more favorable impact of adding ESA to ruxolitinib, with anemia responses occurring in 54% of patients, though the relative timing of ESA initiation to ruxolitinib varied [30].”

  • Reviewer 3: “3) While the review is very updated and also includes works presented at ASH and EHA meetings, the association of ruxolitinib and deferasirox presented at ASH 2019 (Elli E et al. Concomitant Treatment with Ruxolitinib and Deferasirox in the Management of Iron Overload in Patients with Myelofibrosis: A Multicenter Italian Experience. Blood (2019) 134 (Supplement_1): 839. should be reported. I am not sure whether it is more appropriate to include it in the "3. Killing Two Birds with Two Stones – Efforts to Manage Anemia during Ruxolitinib Therapy" (row 86) paragraph or in the "5.15. Other combinations" (row 503) paragraph. I would leave the choiche to the authors.”

Response: This was added to the section on managing anemia during ruxolitinib therapy as iron overload is very closely linked to anemia. We thought it best fit in this section.

The following text has been included to address this comment (Lines 115 – 119), including the study cited as reference 34: “More recently, the impact of iron-chelation therapy (ICT) in combination with ruxolitinib has been assessed. In a retrospective, multi-center study, 59 patients received ruxolitinib in combination with defersirox for the management of iron overload. Iron chelation response was achieved in 24 (41%) of patients and 10 (17%) converted from transfusion dependent to transfusion independent [34].

Round 2

Reviewer 2 Report

The authors very nicely addressed the relevant points. I have no further comments.